

# Characteristics of rainfall events in RCM simulations for the Czech Republic

Vojtěch Svoboda[1], Martin Hanel[1,2], Petr Máca[1], and Jan Kyselý[1,3]

[1]Faculty of Environmental Sciences, Czech University of Life Sciences Prague, Kamýcká 129, Praha 6 – Suchdol, 165 21, Czech Republic
[2]T. G. Masaryk Water Research Institute, Podbabská 2582/30, Praha 6 – Dejvice, 160 00, Czech Republic
[3]Institute of Atmospheric Physics, Czech Academy of Sciences, Boční II 1401, Praha 4 – Spořilov, 141 31, Czech Republic

*Correspondence to:* Vojtěch Svoboda (vsvoboda@fzp.czu.cz)

**Abstract.** Characteristics of rainfall events in an ensemble of 23 regional climate model (RCM) simulations are evaluated against observed data in the Czech Republic for the period 1981–2000. Individual rainfall events are identified using the concept of minimum inter-event time (MIT) and only heavy events during the warm season (May–September) are considered. Inasmuch as an RCM grid box represents a spatial average rather than a point measurement, the effects from areal averaging

of rainfall data on characteristics of events are investigated using the observed data. Rainfall events from the RCM simulations are then compared to those from the at-site and area-average observations. Simulated number of heavy events and seasonal total precipitation due to heavy events are on average represented relatively well despite the higher spatial variation compared to observations. RCM-simulated event depths are comparable to the area-average observations, while event durations are overestimated and other characteristics related to rainfall intensity are significantly underestimated. Most of the rainfall event

characteristics in the majority of the RCM simulations show a similar altitude-dependence pattern as in the observed data. The number of heavy events and seasonal total precipitation due to heavy events increase with altitude, and this dependence is captured better by the RCM simulations with higher spatial resolution.

## 1 Introduction

Not only are such characteristics of heavy rainfall events as event depth, duration, or intensity relevant for urban hydrology (e.g.

Einfalt et al., 1998; Barbosa et al., 2012; Willems et al., 2012) or soil erosion assessment studies (e.g. Angulo-Martínez and Beguería, 2009; Meusburger et al., 2012; Todisco, 2014), they also influence such hydrological processes involving vegetation as interception, throughfall, and stemflow (e.g. Staelens et al., 2008; Siegert and Levia, 2014; Lozano-Parra et al., 2015). Moreover, potential changes in characteristics of precipitation events due to climate change may have significant societal impacts, especially when daily extreme rainfall intensity increases (e.g. Alexander et al., 2006; Trenberth, 2011; Westra et al., 2014).

Short-duration extreme events in particular might intensify more in future climate due to dynamical feedbacks (Lenderink and van Meijgaard, 2008; Berg and Haerter, 2013; Millán, 2014). Despite their importance, however, only a few studies have dealt with the characteristics of individual rainfall events derived from observed rainfall data (Fiener et al., 2013; Hanel et al., 2016). A comparative review of the published rainfall event characteristics has been provided by Dunkerley (2008a).



Many studies of precipitation extremes across Europe examine daily data from regional climate model (RCM) simulations (e.g. Jacob et al., 2007; Fowler and Ekström, 2009; Herrera et al., 2010; Hanel and Buishand, 2011, 2012). Although in recent years growing attention has been given to studies at sub-daily time scales (Hanel and Buishand, 2010; Arnbjerg-Nielsen, 2012; Kendon et al., 2012; Gregersen et al., 2013; Chan et al., 2014a; Kendon et al., 2014; Sunyer et al., 2015), the complexity of

physical processes related to sub-daily extremes (Stocker et al., 2013; Siler and Roe, 2014) and their simplification within climate model parameterizations might discourage researchers from verification of sub-daily simulated precipitation. The lack of long and high-quality observed rainfall data series at hourly or sub-hourly time scales presents another barrier to verification (Westra et al., 2014).

Even as the majority of RCM simulations available are conducted in resolution coarser than 10 km, the convective pro-

cesses associated with extreme rainfall actually develop at much finer scales ($< 4$ km; Prein et al., 2015). RCMs usually rely, therefore, on convection parameterization schemes, even though these are known sources of significant uncertainties and errors (Brockhaus et al., 2008; Hohenegger et al., 2008; Kendon et al., 2012). That is especially the case for the simulated sub-daily precipitation (Westra et al., 2014).

For the Czech Republic and an ensemble of RCMs, Kyselý et al. (2016) reported that underestimation of convective and

overestimation of stratiform precipitation extremes result in a relatively good representation of total daily precipitation extremes. This had been noted also for other regions, including the the Netherlands (Hanel and Buishand, 2010) and Denmark (Gregersen et al., 2013). Moreover, the intensity of convective precipitation and total depths in summer were underestimated (more so in lowlands; Kyselý et al., 2016). Underestimation of summer precipitation totals was reported also by, for example, Hanel and Buishand (2012) (for the Czech Republic) and Rauscher et al. (2010) (for the region Eastern Europe). Kjellström

et al. (2010) indicated that agreement between RCMs and observations at daily time scales was best for moderate rainfall intensities and that bias was increasing for heavier rainfall events. The ability of RCMs to represent hourly rainfall extremes properly has been questioned in several studies (e.g. Hanel and Buishand, 2010; Gregersen et al., 2013; Kendon et al., 2014). For instance, Hanel and Buishand (2010) found that 1 h maxima were underestimated (compared to those from radar data) in most of those RCM simulations which they analysed for the Netherlands.

Characteristics of individual heavy rainfall events (such as event depth, duration, and intensity) in RCM simulations have not been studied to date. Therefore, the purpose of this study is to assess rainfall event characteristics in an ensemble of RCM simulations using hourly data conducted within the ENSEMBLES (van der Linden and Mitchell, 2009) and EURO-CORDEX (Jacob et al., 2014) projects. The simulated rainfall events are compared to area-average observations in the Czech Republic for the validation period 1981–2000.

The paper is organized as follows: Section 2 describes the study area, as well as the observed and RCM-simulated data. Section 3 presents the definition of rainfall events, the event characteristics considered, and the approaches for assessing areal-averaging effects and biases in the RCM simulations. Results concerning the effects from areal averaging of rainfall data on events and evaluation of RCM-simulated rainfall event characteristics are presented in Sect. 4 and discussed in Sect. 5. Key findings are summarized in Sect. 6.





## 2 Study area and data

### 2.1 Study area

Rainfall event characteristics are analysed for the Czech Republic (78,800 $km^2$), located in Central Europe (Fig. 1a). Orography of the country varies considerably. As can be seen in Fig. 1b, approximately two-thirds of the area is situated at altitudes below
500 m a.s.l., even as several mountain ranges exceed 1200 m a.s.l.

Average annual precipitation totals for the 1961–2000 period vary from about 420 mm in the central-western part of the country to more than 1200 mm in the mountains. Mean annual precipitation for the Czech Republic is about 670 mm, with a single maximum occurring at most stations in June and July (Tolasz, 2007). If averaged across the Czech Republic, almost two-thirds of the annual precipitation falls in the warm half of the year. Rainfall events during the warm period (April–September)
are usually of shorter duration and greater intensity. Rainfall or snowfall events during the cold half of the year (October–March), meanwhile, are mainly characterized by lower intensities and longer durations, and these are associated with passing frontal systems and pressure lows (Tolasz, 2007).

### 2.2 Observed precipitation data

In the present study, we used hourly precipitation data provided by the Czech Hydrometeorological Institute. The original data
in 10 min resolution are based on digitized pluviograph records (from float-type self-recording pluviographs with interception area of 250 $cm^2$) and were quality checked by Květoň et al. (2004), who identified and reconstructed damaged or missing pluviograph records while considering many sources of rainfall information. Hanel and Máca (2014) had further assessed the quality of the reconstructed data set by comparing daily precipitation depths aggregated from 10 min data with daily precipitation depths from standard ombrometers. They had considered data for a day unreliable when the difference exceeded
1.5 mm for daily precipitation totals below 15 mm or 10 % for daily precipitation totals above 15 mm. The years with the fraction of unreliable records larger than 10 % were excluded from the data set. These same criteria were applied in the present study.

Given the unreliability of the pluviograph records in the winter period (Květoň et al., 2004), only records from May to September have been considered. This period is hereinafter referred to as the "season". Only stations with at least 10 years
of reliable data were used for the comparison of RCM-simulated and observed rainfall event characteristics in the validation period 1981–2000. The validation period was set to just 20 years in order to maximize overlap of the available RCM simulations (Sect. 2.3) and observed data. In total, 154 stations (with density of 1 station per 512 $km^2$) fulfilled this condition. (Meanwhile, more than 15 years of reliable data were available for 60 stations.)

In order to increase the number of stations available for spatial averaging, a longer period (1961–2009) was considered
for analysing the areal-averaging effects. This resulted in making 26 additional stations available (each of which has records shorter than 10 years or ending before 1981). Figure 1b shows all 180 stations from the data set, the density of which came to approximately 1 station per 438 $km^2$.



Moreover, we examined the influence of the number of stations considered in the areal averaging using a dense rain gauge network for Prague (22 stations within $500 \, \mathrm{km}^2$). Hourly precipitation data for the period 2002–2011 were provided by Pražská vodohospodářská společnost a.s., administrator of the Prague water management property.

### 2.3 RCM simulations

An ensemble of 23 RCM simulations was examined (see Table 1 for an overview). Six RCMs were driven by 14 global climate models (GCMs) to produce 19 simulations in total. Two RCMs (with a total of four runs) were also driven by ERA40 (Uppala et al., 2005) or ERA-INTERIM (Dee et al., 2011) reanalysis.

The RCMs' outputs are available on a rotated latitude–longitude grid with horizontal resolutions ranging from 12.5 to 50 km (Table 1). Only the CLM simulation is on a regular grid. From each RCM simulation only grid boxes covering the area of the Czech Republic (i.e. 52–607 grid boxes for different resolutions and RCMs) were selected.

The HIRHAM5, HadRM3, and RACMO2 simulations were conducted within the ENSEMBLES project (van der Linden and Mitchell, 2009), while the RCA4 and RACMO22E simulations were within the EURO-CORDEX project (Jacob et al., 2014). Two of the HadRM3 simulations were driven by the GCM version with perturbed physics parameterizations (Collins et al., 2006). HadCM3Q0 is an unperturbed model run, HadCM3Q3 is a version with a low sensitivity to external forcing, and HadCM3Q16 includes perturbations resulting in high sensitivity to external forcing. The perturbations in the HadRM3 RCM correspond to those in the HadCM3 GCM.

## 3 Methods

This section defines rainfall events in the observed and RCM-simulated data (Sect. 3.1) and describes those event characteristics considered (Sect. 3.2). Inasmuch as the RCM data represent areal averages rather than point values, the methods for assessing the effect on event characteristics from areal averaging of rainfall data are further described in Sect. 3.3. Finally, approaches taken in evaluating simulated rainfall events are presented in Sect. 3.4.

### 3.1 Rainfall event definition

Several methods exist for defining individual rainfall events (e.g. Peters and Christensen, 2006; Ignaccolo and Michele, 2010; Gaál et al., 2014). One approach frequently used involves the concept of minimum inter-event time (MIT), which defines events on the basis of a minimum time interval – reached or exceeded – between two individual events (Dunkerley, 2008b; Ignaccolo and Michele, 2010). The value of MIT should be selected so that the rainfall events are independent. Nevertheless, MIT is often set on an ad hoc basis, for instance by following another well-established method such as the universal soil loss equation (USLE; Wischmeier and Smith, 1978).

Although the estimated optimal MIT would generally vary between the RCM simulations and across the area, and because the value of MIT influences the values of rainfall events characteristics, a constant MIT is required in order to provide comparable characteristics (e.g. Dunkerley, 2008b; Hanel and Máca, 2014). Therefore, a 6 h MIT was used for deriving rainfall events




within this study for all RCM simulations as well as for the observed data. It should be noted that although this value is lower than the optimal MITs estimated for the Czech Republic by Hanel and Máca (2014), it is one of the values most frequently used (Dunkerley, 2008b). Similarly as in the USLE methodology, we considered further only the heavy rainfall events, i.e. at-site events with total depth greater than 12.7 mm (Wischmeier and Smith, 1978).

5      For the observed data, the minimum wet-hour depth is related to rain gauge precision (0.1 mm). The same wet-hour threshold as used here had been applied also for RCM simulations in previous studies (e.g. Willems and Vrac, 2011; Kendon et al., 2014; Sunyer et al., 2016). Although the same minimum event depth could be taken for RCM simulations as for observations (12.7 mm), it is well known that the RCM-simulated precipitation totals are often biased. As a consequence, the number of selected events might be influenced by the bias in mean precipitation. Alternatively, the minimum event depth of a heavy rainfall event

10    is chosen such that the number of heavy events is on average the same as for observations ($\approx$ 15 %). This approach is similar to the quantile mapping method used frequently for bias correction.

To summarize, the areal heavy rainfall events (in RCM simulations and area-average observations) are here defined while considering a 6 h MIT, fixed wet-hour threshold of 0.1 mm, and minimum event depth corresponding to the 0.85 quantile of the distribution of event depths.

15   ## 3.2   Rainfall event characteristics

We focused on the following basic characteristics of rainfall events:

- event depth $D$ [mm],

- event duration $T$ [h],

- event mean rainfall rate $R$ [mm h$^{-1}$]:

$$R = \frac{D}{T}, \text{ and} \tag{1}$$

- maximum 60 min rainfall intensity during the event $I_{60}$ [mm h$^{-1}$].

As our definition of a rainfall event is in general consistent with the USLE methodology, we consider also indicators of rainfall event erosivity:

- event rainfall energy $E$ [MJ ha$^{-1}$] (Brown and Foster, 1987):

$$E = \sum_{t=1}^{T} 0.29 d_t [1 - 0.72 \exp(-0.05 d_t)], \tag{2}$$

where $d_t$ is rainfall volume during hour $t$, and

- event rainfall erosivity index $EI_{60}$ [MJ mm ha$^{-1}$ h$^{-1}$]:

$$EI_{60} = E \cdot I_{60}. \tag{3}$$





Note that in the USLE methodology, maximum 30 min rainfall intensity is considered. Due to the temporal resolution of RCM-simulated data used in this study, we instead consider maximum 60 min rainfall intensity during an event ($I_{60}$). The $E$ and $EI_{60}$ indices are considered here not in order to quantify soil loss but rather as indicators as to the erosive potential of a rainfall event.

In addition to the aforementioned rainfall event characteristics, we analysed also the following seasonal (May–September) characteristics:

- number of heavy rainfall events per season $N_{se}$ [$-$], and

- seasonal total precipitation due to heavy rainfall events $S_{se}$ [mm].

### 3.3 Areal averaging of rainfall data

Areal averaging of rainfall data can significantly affect such characteristics of the events as depths (Svensson and Jones, 2010) or intensity (Eggert et al., 2015). Because an RCM grid box represents a spatial average, RCM simulations cannot be compared directly to at-site observations. Therefore, various gridded data sets are used for validation of RCM data (e.g. the E-OBS data set for Europe; Haylock et al., 2008). The gridded data sets available are limited to daily or longer temporal resolution. Therefore, analogously to the well-established areal reduction factors describing the decrease in rainfall/runoff maxima with increasing averaging area, we quantified the effect that the areal averaging of rainfall data has on the rainfall event characteristics.

To assess this effect on the (event and seasonal) characteristics, we compared characteristics of events in area-average and at-site observed data as follows:

1. Square neighbourhoods with area corresponding to the considered resolutions (12.5, 25, and 50 km) were defined around each station. Neighbourhoods with only one station were excluded. This resulted in 36 neighbourhoods for the 12.5 km resolution, 118 for the 25 km, and 180 for the 50 km resolution. Average numbers of stations included in the neighbourhoods were 2.25 for the 12.5 km, 2.9 for the 25 km, and 6.04 for the 50 km resolution.

2. Areal average rainfall was calculated for each neighbourhood. Rainfall events were determined for this areal average as well as at-site for the central station. Characteristics of rainfall events were then calculated (hereafter referred to as "area-average" and "at-site" characteristics, respectively).

3. We evaluated the following indices:

   (a) Ratio of areal mean (event and seasonal) characteristics for each neighbourhood to the mean at-site (event and seasonal) characteristics, averaged over all neighbourhoods in the Czech Republic. This ratio is further denoted $rt_{m}$.

   (b) Ratio of the $p^{th}$ quantiles of event characteristics for each neighbourhood to the $p^{th}$ quantiles of at-site event characteristics, averaged across all neighbourhoods in the Czech Republic for $p = 0.05, 0.1, \ldots, 0.95$. This ratio is referred to as quantile ratio $rt_{p}$.



(c) Ratio of seasonal frequencies (averaged across the Czech Republic) for corresponding bins of the histograms of areal and at-site event characteristics, further denoted as histogram ratio $rt_f$.

## 3.4  Evaluation of RCM-simulated characteristics of rainfall events

The RCM-simulated event characteristics (i.e. their areal averages) were compared to the observed at-site characteristics in a manner similar to that described in point 3 of the procedure specified in Sect. 3.3 by replacing the observed areal average (event and seasonal) characteristics for the neighbourhoods with the values for individual grid boxes.

We note that the ratios between the RCM simulations and observed at-site rainfall combine the bias in the RCM simulation with the effect of areal averaging of rainfall data. Therefore, ratios for RCM-simulated characteristics were further compared to those for area-average observations.

Finally, we also examined the dependence of the RCM-simulated (event and seasonal) characteristics on altitude. A linear regression model of the dependence of the $p^{\text{th}}$ quantile (for $p$ from 0.05 to 0.95) of the distribution of event characteristics on altitude was fitted while considering the RCMs and at-site data. Altitude dependence of rainfall event characteristics was then examined as the change of a characteristic per 100 m change in elevation ($y_p$) given by:

$$y_p = \frac{100\beta + \alpha}{\alpha} \cdot 100 \; [\%], \tag{4}$$

where $\beta$ is the slope coefficient and $\alpha$ is the intercept. Moreover, the values of the estimated slope coefficient ($\beta$) were analysed for the seasonal characteristics.

## 4  Results

This section presents findings related to areal averaging of rainfall data (Sect. 4.1). Further, the RCM-simulated rainfall event characteristics are evaluated with respect to the observed data for the validation period (1981–2000) using ratios of mean characteristics ($rt_m$; Sect. 4.2), quantile ratios ($rt_p$; Sect. 4.3), and histogram ratios ($rt_f$; Sect. 4.4). Altitude-dependence of event characteristics considering the RCM simulations and at-site observations is assessed in Sect. 4.5.

### 4.1  Effects of areal averaging in the observed data

The number of rainfall events in area-average observations is higher than that for the at-site data. The same holds true for the number of area-average heavy rainfall events ($N_{se}$) for the three spatial resolutions considered (Table 2, top four rows), which is on average approximately 1.4–2 events per season higher (i.e. by as much as 26 %) than the number of at-site events. Minimum event depth (based on the $85^{\text{th}}$ quantile of event depths) for area-average observations is lower than 12.7 mm (the value for the at-site data), and that results in smaller average event depths ($D$) compared to these for the at-site observations. Smaller $D$ are compensated, however, by a higher number of events ($N_{se}$), and that leads to a good representation of seasonal totals due to heavy rainfall events ($S_{se}$; +3 % for area-average observations on average).





Mean characteristics of rainfall events considered for the at-site and area-average observations are shown in the right part of Table 2. Observed area-average characteristics are in general lower than at-site characteristics: $D$ by 14–19 %, $T$ by 8–18 %, $R$ by 10–27 %, and $I_{60}$ by 23–39 %.

The quantile ratios $rt_\mathrm{p}$ are shown in Fig. 2. Because in general the spread of the quantile ratios is similar for the spatial resolutions considered (not shown), only the envelope of $rt_\mathrm{p}$ for all neighbourhoods representing the maximum range between the 5th and 95th quantiles for the three spatial resolutions is indicated by grey areas in Fig. 2.

Area-average event depths ($D$) are $\approx$ 10–20 % smaller than at-site $D$ for the whole range of event depths. The area-average duration ($T$) of short events is comparable to the at-site $T$ for the 12.5 and 25 km resolutions and larger (by $\approx$ 20 %) for the 50 km resolution. For long events, the area-average $T$ is about 16 % shorter compared to the at-site observations. The area-average rainfall rate ($R$) is similar to the at-site observations for low values and then significantly diminishes with increasing rainfall rate. The remaining area-average characteristics' values are lower than those of the at-site observations, and the differences are larger for higher quantiles.

To demonstrate how spatial resolution influences the area-average characteristics, boxplots of $rt_\mathrm{m}$ for the area-average observations are presented as a part of Fig. 3 (grey boxplots in the right part of each panel). Most of the area-average characteristics decrease with increasing area. The exception is event duration ($T$), which increases with area. The differences in mean event characteristics between the considered spatial resolutions are generally small (less than 10 %), with the exceptions of $R$, $I_{60}$ and $EI_{60}$ (at 17, 16, and 21 %, respectively). These differences are considerably smaller, however, than are the differences between the event characteristics over the study area.

### 4.2 Simulated mean (event and seasonal) characteristics

Figure 3 presents boxplots of $rt_\mathrm{m}$ ratios between the RCM-simulated and observed at-site rainfall event characteristics for the validation period (1981–2000) as derived for grid boxes over the study area.

In the RCM simulations, the event depths ($D$) correspond generally well with the area-average observations (20 % smaller on average compared to at-site observations). Event duration ($T$) is longer for most of the RCM simulations than for at-site observations (on average by 18 %). That is in contrast with the area-average observations, for which the event duration ($T$) is shorter than for at-site observations (by 13 % on average). Because event depths ($D$) for the RCM simulations are smaller than for at-site observations and event durations ($T$) are longer in general, event mean rainfall rates ($R$) are significantly lower compared to the at-site characteristics (by 56 % on average) while $R$ for area-average observations are lower only by 17 % on average. Other characteristics' values, too, are significantly lower for RCM simulations compared to both the area-average and the at-site characteristics. For example, maximum 60 min rainfall intensities during an event ($I_{60}$) are lower by 30 % with respect to the area-average observations and by 69 % on average with respect to the at-site data.

The number of heavy events per season ($N_\mathrm{se}$) is in general higher in the RCM simulations compared to the at-site observations (by 42 % on average, i.e. 3.2 events per season). That difference is only 16 % (less than 1.5 events per season), however, when compared to the area-average observations. The differences in $S_\mathrm{se}$ between the RCM simulations and both at-site and area-average observations range from $-33$ to $+48$ % ($+11$ % on average, i.e. 23 mm per season compared to the area-average



observations). For the two HIRHAM5 simulations (H5_BCM and H5_ECHAM5), large outliers are present at the grid boxes with the highest altitude (in the northern part of the Czech Republic). There, the number of simulated heavy events ($N_{se}$) reaches to approximately 44 per season and the seasonal total precipitation due to heavy events ($S_{se}$) to approximately 1700 mm. For observations (both at-site and area-average), maximum values are given by stations at highest altitudes, as well, but

the maximum $N_{se}$ is 16 and that for $S_{se}$ is $\approx 500$ mm. Moreover, several other outliers with higher values for the characteristics are present in the RCM simulations (especially for event depths and energies; not shown in Fig. 3). These outliers, too, are generally linked to grid boxes with the highest altitudes.

The coefficient of variation (CV; not shown) of $rt_m$ for the RCM grid boxes (indicator of spatial variability) corresponds relatively well with that of the area-average observations for event depth ($D$; CV about 9 % on average), duration ($T$; about

12 %), and kinetic energy ($E$; about 10 %). The RCM-simulated spatial variability is somewhat lower compared to the area-average observations for the other characteristics (CV is 9 % for $R$, 8 % for $I_{60}$, and 18 % for $EI_{60}$ for the RCM simulations whereas these are 12 %, 12 %, and 29 %, respectively, for the area-average observations). CV for the number of heavy events per season ($N_{se}$) and seasonal totals due to heavy events ($S_{se}$) is significantly higher in most of the RCM simulations (by almost as much as three times for $N_{se}$ in several RCA4 simulations). Only for RACMO2 and RACMO22E simulations do CVs for

$N_{se}$ and $S_{se}$ correspond well with the area-average observations (CV of about 25 and 30 % for $N_{se}$ and $S_{se}$, respectively).

## 4.3 Quantile ratios ($rt_p$)

Figure 4 gives the quantile ratios ($rt_p$) for the RCM simulations and the area-average observations. For the latter, only the average value (from the 12.5, 25, and 50 km neighbourhoods) and envelope representing the maximum 5th–95th quantile range for all neighbourhoods is shown (for details, see Sect. 4.1 and Fig. 2).

The correspondence between simulated and area-average event depths ($D$) is best for events with large $D$. Some of the RCM simulations underestimate depths ($D$) for events with low $D$ (which results in 7 % underestimation on average of $D$ for the 0.05 quantile in the RCM simulations). For most of the RCA4 runs, the quantile ratio $rt_p$ does not depend on event depth ($D$). That is also the case for area-average observations.

For the longest events, the RCM-simulated event duration ($T$) corresponds relatively well with the area-average and at-

site observations. However, shortest durations ($T$) are greatly overestimated (by 1.5 to 3.1 times; i.e. 2.3 times in the RCM simulations on average compared to the area-average characteristics). Only the event duration ($T$) in the H5_ARPEGE matches that of the area-average observations for the whole range of event durations.

The difference between areal (RCM simulations as well as area-average observations) and at-site rainfall rate ($R$) grows with increasing quantile. Stronger underestimation of largest rainfall rates ($R$) is related to significant overestimation of shortest

durations ($T$). Moreover, underestimation of the maximum 60 min intensities ($I_{60}$) is greater for larger values.

Event rainfall energy ($E$) is related to the values for event depth ($D$) and duration ($T$). The overestimation of duration ($T$) for short events partly compensates for the underestimation of small depths ($D$), and that results in 15 % underestimation of the rainfall energy ($E$) in the RCM simulations for low quantiles. As quantile increases the differences increase as well. Due





to good correspondence between simulated and observed area-average largest event depths ($D$), however, the underestimation of $E$ is smaller for the highest quantiles.

### 4.4 Histogram ratios ($rt_f$)

Differences in distribution of rainfall event characteristics' values between the RCM simulations and the at-site observations are described by histogram ratio $rt_f$ (Fig. 5).

Considerably higher numbers of events with smaller depths ($D$) in the RCM simulations compared to the at-site observations are expected, partly due to the definition of a rainfall event (minimum values of $D$ in the RCM simulations are between 7 and 11.5 mm). Approximately 2.5 % of all considered events for at-site observations have depths ($D$) smaller than 13 mm, while for area-average observations and the three spatial resolutions considered the proportion of such events is 22–29 %. For the ensemble of RCM simulations, this range takes in 15–54 % of events. Extreme heavy events (with depths exceeding 250 mm), which rarely occur at some stations, are not present in most of the RCM simulations (exceptions being for all HIRHAM5 runs, HadRM3Q16 driven by ERA40, and HadRM3 driven by HadCM3).

Simulated numbers of events with short duration ($T$) are underestimated. Only 0.3–7.5 % (1.6 % on average) of events considered for the RCM simulations are shorter than 6 h, while for the area-average observations it is 9–13 % of events (15 % for the at-site observations).

Events with the smallest rainfall rates ($R < 0.5 \, \mathrm{mm \, h^{-1}}$) are more frequent in the RCM simulations (8–28 % of considered events) than in the area-average (5–8 %) and at-site observations (7 %). On the other hand, high rainfall rates ($R > 3 \, \mathrm{mm \, h^{-1}}$) are very rare for the RCM simulations (0.3–4 % of considered events), while for the area-average observations these represent from 8 to 15 % (and for at-site observations almost 19 %) of considered events.

Most of the simulated events (84–99 %) have maximum 60 min intensity ($I_{60}$) less than 6 $\mathrm{mm \, h^{-1}}$. Among area-average and at-site observations, meanwhile, just 60–72 % and 40 % of events, respectively, have $I_{60} < 6 \, \mathrm{mm \, h^{-1}}$. RCA4 simulations (50 km resolution) have absolute maximum values of $I_{60}$ significantly lower compared to those of other RCM simulations (only around 11 $\mathrm{mm \, h^{-1}}$).

Only 24 % of the observed at-site events considered have rainfall energy ($E$) below 2 $\mathrm{MJ \, ha^{-1}}$, whereas for the area-average observations this figure is 51–58 % and for the RCM simulations it is 38–62 % (with mean 54 %) of events. The RCM simulations also have more events with low rainfall erosivity index ($EI_{60}$). For the RCM simulations, 68–92 % (with mean of 83 %) of events have $EI_{60} < 10 \, \mathrm{MJ \, mm \, ha^{-1} \, h^{-1}}$, and for the area-average observations that range is 50–60 %. For the at-site observations, this low erosivity index occurs for only 26 % of events.

### 4.5 Altitude-dependence

Figure 6 shows the altitude-dependence of event characteristics for the RCM simulations and at-site observations (note that for the area-average observations the altitude-dependence has not been investigated due to an uneven spatial distribution and different numbers of stations in neighbourhoods). The altitude-dependence is expressed as the change in characteristics values





per 100 m in elevation as estimated by a linear regression between altitude and the values of the rainfall event characteristics for a specific quantile.

Although the RCM simulations generally show a similar pattern of altitude-dependence as that for the at-site observations regarding most characteristics (with trends between $-5$ and 10 %/100 m for all quantiles), several RCM simulations at 25 and 50 km resolutions show stronger altitude-dependence compared to at-site observations for high (event depth, duration, rainfall energy, and rainfall erosivity index) or low (rainfall rate and maximum 60 min intensity) quantiles of rainfall event characteristics. Two simulations with the highest horizontal resolution (RACMO22E and CLM) show different behaviour for greatest event depths ($D$) compared to those of other RCM simulations and at-site observations inasmuch as large $D$ does not increase with altitude. These differences are nevertheless less than 5 %.

The linear regression is significant (with $p < 0.05$) for 78 % of all assessed quantiles for all characteristics and the at-site observations and for 68 % of those quantiles in the case of the RCM simulations. For RCM-simulated rainfall rates ($R$) and maximum 60 min intensities ($I_{60}$), however, the altitude-dependence is found to be significant much less frequently (58 and 51 % of cases) than for at-site observations (100 and 95 %, respectively). The number of significant regressions was also considerably smaller for the RCA4 runs with 50 km resolution.

Number of heavy events per season ($N_{se}$) and seasonal totals due to heavy events ($S_{se}$) strongly depend on altitude in the RCM simulations (not shown). Simulated $N_{se}$ increase with altitude, with a slope coefficient $\beta$ 1.8–7.7 times greater than those for at-site observations. A similar situation is found for $S_{se}$, with $\beta$ 1.1–5.9 times greater. These steeper slope coefficients $\beta$ make altitude-dependence of seasonal characteristics unrealistic compared to at-site observations for a large part of the RCM simulations. Only two RCM simulations with the higher spatial resolution (CLM with 11.7 and 13.9 %/100 m, and RACMO22E with 15 and 21 %/100 m) represent the altitude-dependence of $N_{se}$ and $S_{se}$ for the Czech Republic adequately (trends for at-site observations are 8.4 and 13.4 %/100 m, respectively).

## 5 Discussion

### 5.1 Definition and characteristics of rainfall events

Heavy rainfall event characteristics were assessed in an ensemble of 23 RCM simulations. Events were identified while considering 6 h minimum inter-event time (MIT), 0.1 mm fixed wet-hour threshold, and minimum total event depth derived for each RCM simulation as event depth with the same exceedance probability as for the 12.7 mm depth in the observed at-site rainfall events. Setting the minimum event depth to the fixed value of 12.7 mm led to results generally similar to those presented in Sect. 4. On the other hand, setting the wet-hour threshold using quantile matching (i.e. in general to lower than 0.1 mm) resulted in excessive prolongation of rainfall events in some RCM simulations and distortion in the distributions of other event characteristics.

While the 6 h MIT was used here for the sake of consistency with the definition in other studies (e.g. Agnese et al., 2006; Murakami, 2006; Fiener et al., 2013; Hanel et al., 2016) and given by the USLE, the optimal MIT estimated by Hanel and Máca (2014) for the Czech Republic was considerably larger (426–2055 min, 763 min on average). Larger values of optimal





MIT had been reported also by Dunkerley (2008b). To explore the optimal MIT for the RCM simulations, we used the same procedure as considered by Hanel and Máca (2014) and which is based on the coefficient of variation (CV) of the inter-event times (Restrepo-Posada and Eagleson, 1982). Optimal MIT averaged across grid boxes for the RCM simulations varied from 7 h (RCA4_NorESM1-M) to 27 h (HadRM3Q3_ HadCM3), with an average value of $\approx 13$ h. Differences in the optimal MIT

between grid boxes for the individual RCM simulations were large, ranging from 4 h for RCA4_CM5A-MR (optimal MIT in the range of 8–11 h) to more than 24 h for H5_ARPEGE (14–41 h). The spatial variability was greatest for the three RCM simulations with the highest optimal MIT: HadRM3Q3_HadCM3, HadRM3Q3_ERA40, and H5_ARPEGE. Because MIT value strongly influences the values of event characteristics (e.g. higher MIT leads to a smaller number of events, consequently with larger duration and event depths; Hanel and Máca, 2014), the evaluation of event characteristics would not be feasible

using estimated, highly variable MITs.

## 5.2 Effect of areal averaging

The areal-averaging effect was considered when comparing at-site observed and RCM-simulated event characteristics. Observed area-average rainfall event characteristics were analysed for an extended period (1961–2009) in three spatial resolutions (12.5, 25, and 50 km) corresponding to the resolutions of the RCMs.

Estimates as to the effect of areal averaging are influenced by several sources of uncertainties (Svensson and Jones, 2010), in particular by spatial variability and coverage of rain gauges. Because it is obvious that a sufficient number of stations must be available in order to provide a reliable estimate of the areal-averaging effect, we assessed the effect of the number of stations considered in the areal averaging on the estimated $rt_m$ ratio using a dense rain gauge network for Prague (22 stations, 500 km$^2$). The assessment was based on repetitive estimation of $rt_m$ from resampled data with the number of stations used for

areal averaging ranging from 2 to 22.

The number of stations included into the calculation of the areal average influenced the estimated event characteristics. Mean event depths ($D$) were least affected by the number of stations compared to other characteristics (with $rt_m$ decreasing with the number of stations from $-12$ to $-15$ % of the at-site value). Event duration ($T$) was slightly increasing with the number of stations involved (from $-5.6$ to $4.3$ %). Behaviour of $D$ and $T$ was reflected in the event mean rainfall rate ($R$), which was

continually decreasing with the rising number of stations from $-17$ to $-39$ %. As a consequence, the ratios for maximum 60 min intensity ($I_{60}$; $-17$ to $-33$ %), event rainfall energy ($E$; $-18$ to $-26$ %), and event rainfall erosivity index ($EI_{60}$; $-36$ to $-59$ %) were decreasing with the number of stations as well. The largest differences in ratios were observed between areal averages estimated from a small number of stations (typically fewer than 6). This finding is in agreement with Allen and DeGaetano (2005), who reported that areal reduction factors are not substantially influenced by the number of stations

involved when derived from 10 or more stations. Observed area-average rainfall event characteristics for the study area (the Czech Republic) can therefore be affected by insufficient number of stations for resolution finer than 50 km (below 3 stations per neighbourhood).

Several conclusions can be drawn from the comparison of area-average and at-site characteristics in general:



- More heavy rainfall events are identified in area-average observations while the area-average seasonal total precipitation due to heavy events corresponds well with that from the at-site observations.

- Area-average event characteristic values are on average lower than are those for at-site observed characteristics, except that area-average event duration is longer for the shortest events and rainfall rate is comparable for events with low rates.

- For most of the characteristics, the difference between the area-average and at-site observations grows with increasing non-exceedance probability (the exception being event depth, for which the difference is comparable across the whole distribution). Many other studies point out larger differences between area-average and at-site rainfall for more extreme rainfall events (e.g. Skaugen, 1997; Asquith and Famiglietti, 2000; Allen and DeGaetano, 2005).

- Considerably fewer events with high maximum 60 min rainfall intensity ($I_{60}$) and more events with low $I_{60}$ occur in area-average observations than in at-site observations.

- The effect of areal averaging (lower values of characteristics with larger area, except for event duration) is generally in agreement with the review published by Svensson and Jones (2010) and the analysis of Eggert et al. (2015), who have shown more pronounced decrease for more extreme convective precipitation intensity with coarser spatial and larger temporal resolution derived from radar data. However, the estimated areal-averaging effect was not much different for the considered area sizes, and especially with respect to its great spatial variability. This might be a consequence of a small number of stations being available for estimation at finer spatial scales.

### 5.3 RCM-simulated rainfall event characteristics

Differences between the RCM-simulated and at-site observed characteristics are in general considerably larger than are those between the at-site and area-average observations, i.e. these differences are dominated by the RCMs' bias rather than the areal-averaging effect.

The ensemble-average number of heavy rainfall events is larger by 1.5 events (16 %) per season than for area-average observations, and the ensemble average of seasonal total precipitation due to heavy events is larger by 23 mm (11 %). Although the RCM-simulated number of heavy events and seasonal total precipitation due to heavy events averaged across the Czech Republic correspond relatively well with the area-average observations, large differences between individual grid boxes may be found (especially due to elevation). Generally good simulation of extremes (mean annual maxima, 20-year return values) in total precipitation amounts (from both convective and stratiform daily precipitation data together) were also reported for the Czech Republic by Kyselý et al. (2016).

Simulated event depths ($D$) are comparable with the area-average observations, with depths of extreme heavy rainfall events (with non-exceedance probability $p = 0.95$) corresponding best with the area-average observations. Because mean simulated (and area-average) event durations ($T$) are relatively long ($> 16$ h), the results are in general consistent with those of Hanel and Buishand (2010), who reported good representation of daily precipitation extremes for the area of the Netherlands even as hourly maxima tended to be too low. For the Czech Republic, Hanel and Buishand (2012) found larger negative bias in daily





precipitation extremes for an ensemble of RCM simulations from the ENSEMBLES project in summer (by as much as 17 %), while the bias was significantly lower in spring and autumn. Hence, including May and September into rainfall data might reduce the bias, which is smaller in our study.

Recent studies considering different spatial resolutions of RCM simulations suggest that hourly precipitation characteristics of extreme events are represented better in RCMs with higher spatial resolution (e.g. Ban et al., 2014; Chan et al., 2014b; Kendon et al., 2014). The better representation of hourly extremes is mainly due to the convection-resolving approach, however, because by increasing spatial resolution to approach the convection-permitting scale (about 4 km; Prein et al., 2015) it is possible to switch off most of the convection parameterizations (Fosser et al., 2015). All RCM simulations analysed in this paper rely on convection parameterization schemes, and the differences between RCM simulations conducted at different spatial resolutions are small and not systematic. Also, Sunyer et al. (2016) suggest that differences between RCM simulations with convective parameterization due to spatial resolution are smaller are than differences due to RCM–GCM combinations for precipitation extremes. Therefore, characteristics of rainfall events for the RCM simulations with 50 km resolution (RCA4) also are not much different from these for other RCM simulations. Exceptions are the highest maximum 60 min rainfall intensities during an event ($I_{60}$), which are underestimated more in RCA4 simulations compared to these from other RCMs. These results are in accordance with those of Sunyer et al. (2016), who concluded that the 50 km spatial resolution is not sufficient to reproduce hourly extreme precipitation even though the performance of the RCMs considering daily extremes seems not to depend on the spatial resolution.

The RCM-simulated maximum 60 min rainfall intensities ($I_{60}$) as well as the number of events with large rainfall rate ($R$) are significantly lower than for the observed data. This is in accord with Kyselý et al. (2016), who concluded that intensity of RCM-simulated convective precipitation in summer is underestimated and that this underestimation is related to convective parameterization. Similar explanation for the underrepresentation of short-duration, high-intensity events in RCM simulations was given also by Kendon et al. (2012).

Overestimation of event duration ($T$) is a consequence of the well-known tendency of RCMs to produce too much persistent light rain and underestimate the number of dry days (e.g. Fowler et al., 2007; Boberg et al., 2009; Kendon et al., 2012). This strongly impacts especially events with the shortest durations ($T$), which are severely overestimated in the RCM simulations. Large overestimation of short durations ($T$) then causes pronounced underestimation of larger rainfall rates ($R$). Fosser et al. (2015) reported the same issue of too-long events also for an RCM with higher resolution and most of the convective parameterizations turned off (COSMO-CLM, 2.8 km resolution).

It should be noted that when event duration ($T$) is corrected (i.e. the events are proportionally shortened according to quantile ratio $rt_p$ of event durations), mean rainfall rate ($R$) increases to values that are well comparable with the area-average observations. For instance, correcting the event durations ($T$) in the RACMO2_ECHAM5 simulation (corresponding best with the area-average event depths) results in an 80 % increase of mean $R$ (mean characteristics are shown in Table 2). Shortening an event also increases the maximum 60 min intensity during an event ($I_{60}$; by about 50 %), event rainfall energy ($E$; by 10 %), and erosivity index ($EI_{60}$; by 65 %), even though these values are still slightly below the area-average observations (ratios $rt_m$ in Fig. 3 for corrected mean $I_{60}$, $E$, and $EI_{60}$ would then be 0.46, 0.61, and 0.26, respectively).





Most analysed characteristics in most of the RCM simulations show a pattern of altitude-dependence similar to that for the at-site observations, and the differences in strength of the altitude-dependence for different quantiles of rainfall event characteristics are in general small (largest differences compared to at-site observations appear for simulations with the coarse 50 km resolution). The number of heavy events per season and seasonal total precipitation due to heavy events increase with

altitude, and this dependence is captured better by RCM simulations with the higher spatial resolution. This could be expected due to better representation of orography as indicated by Rauscher et al. (2010) or Prein et al. (2016).

RCM simulations driven by the ERA reanalyses do not in general show better results compared to those from the GCM-driven RCMs. That is in agreement with Hanel and Buishand (2010), who indicated that bias is largely due to the precipitation parameterization rather than the driving boundary conditions. Although HadRM3 runs driven by the ERA40 reanalysis have

event durations ($T$) considerably shorter than these for the GCM-driven simulations (i.e. they show greater similarity with observations), values for event depths ($D$) as well as other characteristics are generally lower in the ERA40-driven simulations. As a result, the distribution functions of the other event characteristics appear similar for the simulations driven by ERA40 as for those driven by GCMs, with larger differences occurring mostly between the perturbed and unperturbed runs. RCA4 driven by ERA-INTERIM reanalysis remains approximately in the middle of the range of all RCA4 runs for all characteristics and all

assessed ratios. That is in line with the inter-comparison of RCA4 simulations according to monthly precipitation amounts as reported by Strandberg et al. (2014).

## 6   Conclusions

This study presents a methodology for analysis of precipitation characteristics in RCM simulations from an event-based perspective. Individual rainfall events are important with respect to many hydrological applications. Although it is generally not

expected that the current RCMs would simulate sub-daily variability and rainfall event characteristics properly (e.g. Kendon et al., 2014; Westra et al., 2014), characterization of the biases can be useful for studies using simulated sub-daily rainfall data and also for the development of climate models, including research concerning their parameterizations, which is still very pertinent (e.g. Grell and Freitas, 2014) despite the increasing availability of convection-permitting RCM simulations (Prein et al., 2015).

The results suggest that representation of individual rainfall events in the RCM simulations suffers from several deficiencies which have been only partly discussed in previous studies dealing with precipitation characteristics and extremes. The most important findings are summarized as follows:

- Differences between RCM-simulated and at-site observed rainfall event characteristics are dominated by the bias of the climate models rather than the areal-averaging effect.

- The RCMs on average represent the number of heavy rainfall events and seasonal total precipitation due to heavy events relatively well, except that grid boxes at the highest altitudes produce large overestimation for some RCM simulations



(in general, slightly higher numbers of events for most of the RCM simulations; as well as a good representation of seasonal total precipitation for an ensemble of RCMs, despite larger differences for different RCM simulations).

- Simulated event depths correspond relatively well with the area-average observations, while event durations are overestimated. Other characteristics (event mean rainfall rate, maximum 60 min rainfall intensity, and indicators of rainfall event erosivity) are significantly underestimated.

- The underestimation is larger for larger rainfall rates and maximum 60 min rainfall intensities during an event. These characteristics are underestimated most for extreme heavy events.

- The largest deficiencies are found for events with short duration, which are longer in the RCM simulations compared to the area-average observations. Therefore, the numbers of events with shortest duration (below 10 h) also are much lower in the RCM data. Overestimation of event durations then causes underestimation of rainfall rates and partly also of other characteristics.

The limitations in RCM-simulated rainfall event characteristics should be taken into consideration when applying their outputs in hydrological studies and climate change assessments.

*Acknowledgements.* The research was supported by the Czech Science Foundation (project number 14-18675S) and the Ministry of the Interior of the Czech Republic (project number VG20122015092). We thank E. Buonomo (MOHC), O. B. Christensen (DMI), E. van Meijgaard (KNMI), and G. Nikulin (SMHI) for providing the sub-daily RCM data. We acknowledge the ENSEMBLES project, funded by the European Commission's sixth Framework Programme through contract GOCE-CT-2003-505539, and the World Climate Research Programme's Working Group on Regional Climate, and the Working Group on Coupled Modelling, former coordinating body of CORDEX and responsible panel for CMIP5. We also thank the climate modelling groups and institutions (listed in Table 1 of this paper) for producing and making available their model output. We thank the Czech Hydrometeorological Institute and Pražská vodohospodářská společnost a.s. for providing observed at-site data.



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



**Table 1.** RCM simulations analysed.

| **RCM** (reference) – Institution producing the model output | | | |
|---|---|---|---|
| Acronym | Driven by | Horizontal resolution | Number of gridboxes |
| **CLM 2.4.11** (Böhm et al., 2006; Lautenschlager et al., 2009) – Max Planck Institute for Meteorology (MPI), Germany | | | |
| CLM | ECHAM5/MPIOM | 22 km (0.2°) | 301 |
| **HadRM3.0** (Collins et al., 2011) – Met Office Hadley Centre (MOHC), UK | | | |
| HadRM3Q0_ERA40 | ERA40 | 25 km (0.22°) | 173 |
| HadRM3Q3_ERA40 | ERA40 | | |
| HadRM3Q16_ERA40 | ERA40 | | |
| HadRM3Q0_HadCM3 | HadCM3Q0 | | |
| HadRM3Q3_HadCM3 | HadCM3Q3 | | |
| HadRM3Q16_HadCM3 | HadCM3Q16 | | |
| **HIRHAM5** (Christensen et al., 2007) – Danish Meteorological Institute (DMI) | | | |
| H5_ARPEGE | ARPEGE | 25 km (0.22°) | 173 |
| H5_BCM | BCM | | |
| H5_ECHAM5 | ECHAM5-r3 | | |
| **RACMO2.1** (van Meijgaard et al., 2008) – Royal Netherlands Meteorological Institute (KNMI) | | | |
| RACMO2_ECHAM5 | ECHAM5-r3 | 25 km (0.22°) | 173 |
| RACMO2_MIROC | MIROC3.2 | | |
| **RACMO22E** (van Meijgaard et al., 2012) – Royal Netherlands Meteorological Institute (KNMI) | | | |
| RACMO22E | ICHEC-EC-EARTH | 12.5 km (0.11°) | 607 |
| **RCA4.0** (Kupiainen et al., 2011; Samuelsson et al., 2011) – Swedish Meteorological and Hydrological Institute (SMHI) | | | |
| RCA4_ERAINT | ERA-INTERIM | 50 km (0.44°) | 52 |
| RCA4_CanESM2 | CCCma-CanESM2 | | |
| RCA4_CM5A-MR | IPSL-CM5A-MR | | |
| RCA4_CNRM-CM5 | CNRM-CM5 | | |
| RCA4_EC-EARTH | ICHEC-EC-EARTH | | |
| RCA4_ESM2M | NOAA-GFDL-ESM2M | | |
| RCA4_ESM-LR | MPI-ESM-LR | | |
| RCA4_HadGEM2-ES | MOHC-HadGEM2-ES | | |
| RCA4_MIROC5 | MIROC5 | | |
| RCA4_NorESM1-M | NCC-NorESM1-M | | |





**Table 2.** Seasonal number of heavy rainfall events ($N_{se}$ [−]), seasonal total precipitation due to heavy events ($S_{se}$ [mm]), and mean rainfall event characteristics ($D$ [mm], $T$ [mm], $R$ [mm h$^{-1}$], $I_{60}$ [mm h$^{-1}$], $E$ [MJ ha$^{-1}$], and $EI_{60}$ [MJ mm ha$^{-1}$ h$^{-1}$]) for the 1981–2000 period in the observed data (top 4 rows) and the RCM simulations.

| Acronym | $N_{se}$ | $S_{se}$ | $D$ | $T$ | $R$ | $I_{60}$ | $E$ | $EI_{60}$ |
|---|---|---|---|---|---|---|---|---|
| **At-site observations average** | 7.62 | 194.7 | 25.0 | 19.7 | 2.37 | 9.38 | 3.66 | 48.75 |
| **Area-average observations** | | | | | | | | |
| 12.5 km | 9.41 | 205.6 | 21.4 | 16.2 | 2.14 | 7.20 | 2.92 | 28.88 |
| 25 km | 9.00 | 197.9 | 21.6 | 17.1 | 2.02 | 6.82 | 2.89 | 26.16 |
| 50 km | 9.66 | 198.8 | 20.2 | 18.1 | 1.73 | 5.68 | 2.56 | 18.77 |
| **Summary of RCM simulations** | | | | | | | | |
| RCMs average | 10.82 | 223.4 | 20.0 | 23.2 | 1.04 | 2.93 | 2.02 | 7.88 |
| 12.5 km res. RCM | 11.20 | 212.5 | 18.8 | 25.9 | 0.86 | 3.00 | 1.90 | 7.96 |
| 25 km res. RCMs | 10.54 | 215.2 | 19.8 | 22.3 | 1.08 | 3.12 | 2.04 | 9.08 |
| 50 km res. RCMs | 11.08 | 235.0 | 20.5 | 23.9 | 1.02 | 2.60 | 2.01 | 5.97 |
| **RCM simulations** | | | | | | | | |
| CLM | 10.84 | 208.9 | 19.2 | 22.8 | 1.06 | 3.91 | 2.09 | 13.58 |
| HadRM3Q0_ERA40 | 11.83 | 239.2 | 19.6 | 20.6 | 1.19 | 3.74 | 2.11 | 12.28 |
| HadRM3Q3_ERA40 | 10.05 | 169.5 | 16.5 | 18.8 | 1.00 | 2.90 | 1.70 | 6.99 |
| HadRM3Q16_ERA40 | 11.40 | 226.7 | 19.1 | 21.5 | 1.08 | 3.21 | 1.98 | 9.56 |
| HadRM3Q0_HadCM3 | 11.33 | 278.3 | 23.9 | 23.2 | 1.28 | 4.20 | 2.62 | 16.90 |
| HadRM3Q3_HadCM3 | 10.51 | 209.0 | 19.5 | 21.6 | 1.04 | 3.01 | 2.01 | 8.56 |
| HadRM3Q16_HadCM3 | 10.31 | 231.7 | 21.6 | 23.2 | 1.11 | 3.48 | 2.27 | 11.84 |
| H5_ARPEGE | 7.61 | 132.7 | 16.9 | 17.4 | 1.19 | 2.83 | 1.70 | 5.67 |
| H5_BCM | 12.70 | 246.6 | 18.5 | 20.5 | 1.08 | 2.51 | 1.83 | 5.27 |
| H5_ECHAM5 | 11.11 | 224.7 | 19.0 | 20.5 | 1.13 | 2.91 | 1.92 | 6.54 |
| RACMO2_ECHAM5 | 9.42 | 191.0 | 20.2 | 26.6 | 0.91 | 2.81 | 2.02 | 7.69 |
| RACMO2_MIROC | 9.67 | 217.8 | 22.4 | 31.1 | 0.85 | 2.74 | 2.24 | 8.63 |
| RACMO22E | 11.20 | 212.5 | 18.8 | 25.9 | 0.86 | 3.00 | 1.90 | 7.96 |
| RCA4_CanESM2 | 10.65 | 221.5 | 20.0 | 23.9 | 1.01 | 2.63 | 1.97 | 6.01 |
| RCA4_CM5A-MR | 11.14 | 219.4 | 19.2 | 22.7 | 1.02 | 2.53 | 1.89 | 5.46 |
| RCA4_CNRM-CM5 | 12.38 | 289.2 | 22.5 | 25.0 | 1.07 | 2.77 | 2.23 | 6.94 |
| RCA4_EC-EARTH | 11.11 | 249.7 | 21.7 | 26.4 | 0.95 | 2.44 | 2.11 | 5.86 |
| RCA4_ERAINT | 10.20 | 208.1 | 19.9 | 23.0 | 1.03 | 2.67 | 1.97 | 6.04 |
| RCA4_ESM2M | 12.08 | 275.6 | 22.2 | 26.1 | 0.99 | 2.53 | 2.17 | 6.25 |
| RCA4_ESM-LR | 11.45 | 257.9 | 21.9 | 25.0 | 1.06 | 2.72 | 2.17 | 6.65 |
| RCA4_HadGEM2-ES | 9.93 | 173.3 | 16.8 | 21.6 | 0.93 | 2.35 | 1.62 | 4.35 |
| RCA4_MIROC5 | 10.11 | 221.2 | 21.2 | 23.0 | 1.11 | 2.83 | 2.11 | 6.67 |
| RCA4_NorESM1-M | 11.76 | 233.8 | 19.2 | 21.9 | 1.03 | 2.57 | 1.89 | 5.41 |





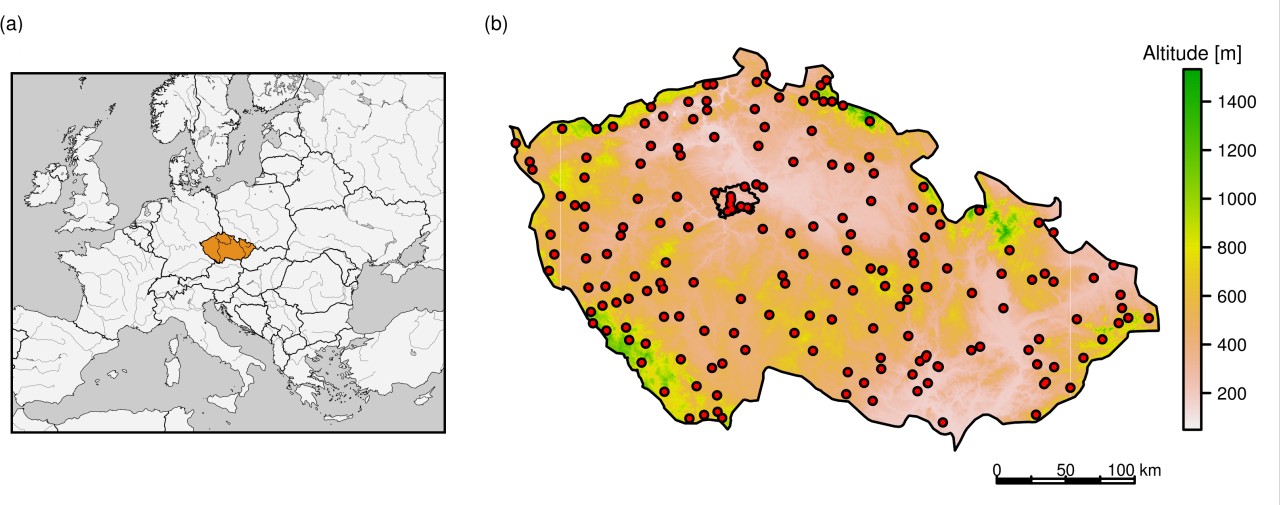

**Figure 1.** (a) Location of the Czech Republic in Central Europe. (b) Stations (dots) used for assessing the areal-averaging effect and evaluating the RCM simulations. Altitude is indicated by colour.

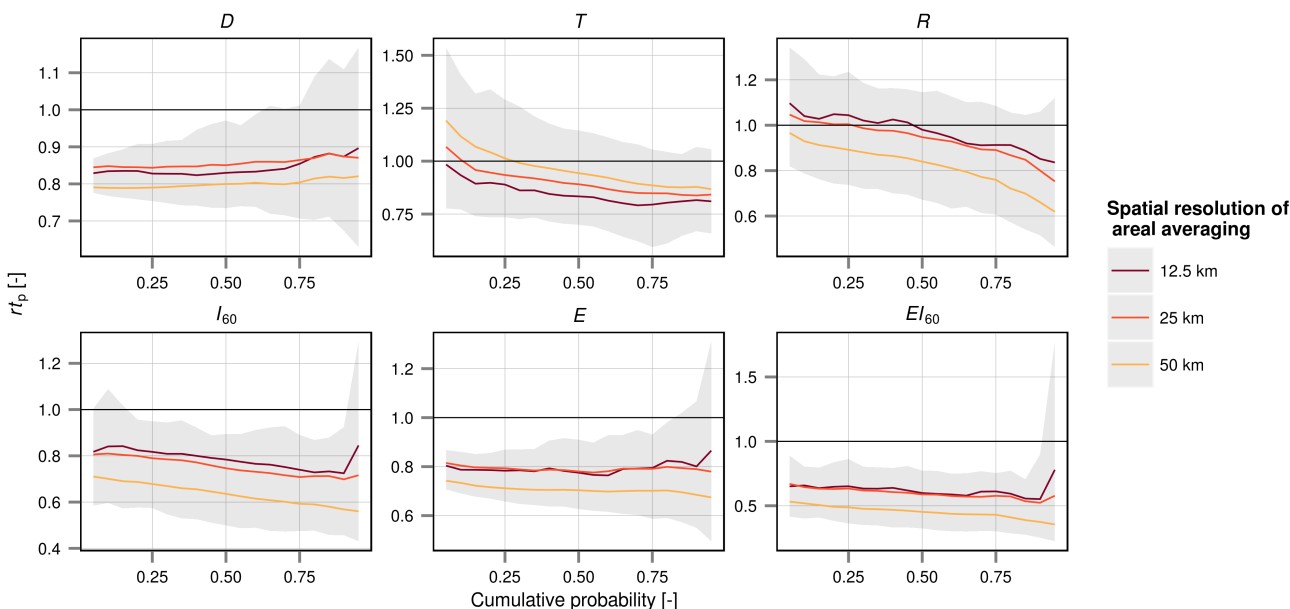

**Figure 2.** Quantile ratios $rt_\mathrm{p}$ for the distribution of the area-average and at-site rainfall event characteristics ($D$, $T$, $R$, $I_{60}$, $E$, $EI_{60}$). For each spatial resolution (12.5, 25, 50 km), the average from all neighbourhoods is indicated by solid line. Envelopes of $rt_\mathrm{p}$ for all neighbourhoods (grey areas) represent the maximum range between the 5[th] and 95[th] quantiles for the three considered spatial resolutions.







**Figure 3.** Ratios $rt_m$ between areal (grid box or neighbourhood) and station average of mean (event and seasonal) characteristics for each RCM simulation and area-average observations (grey boxplots in the right part of each panel). Each boxplot corresponds to values for all grid boxes (neighbourhoods) over the Czech Republic. Grey areas indicate the range between the minimum 25th and maximum 75th quantiles of $rt_m$ for observed data from all neighbourhoods.



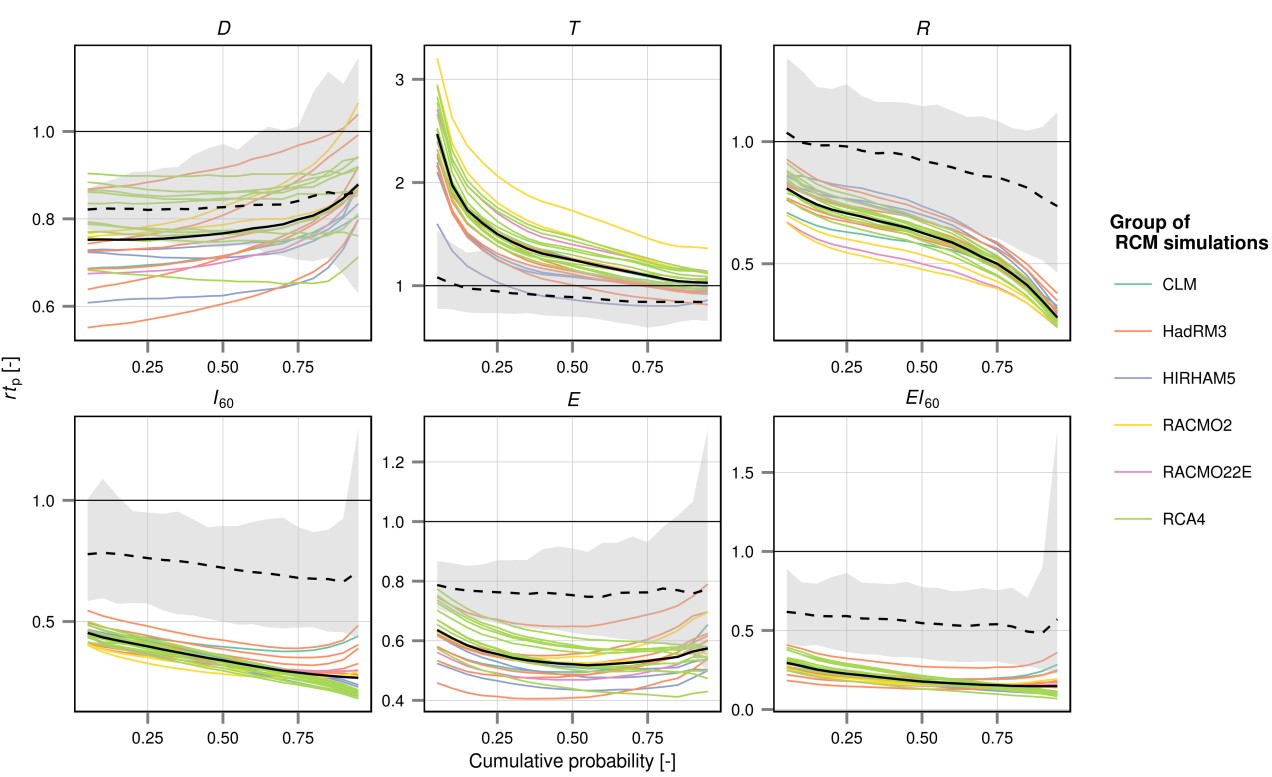

**Figure 4.** Area-average quantile ratios $rt_p$ for the event characteristics derived from RCM simulations. Bold black line shows average from all RCM simulations. Grey area corresponds to the 90 % envelope of the quantile ratios for the area-average observations, and broken line indicates the areal average.





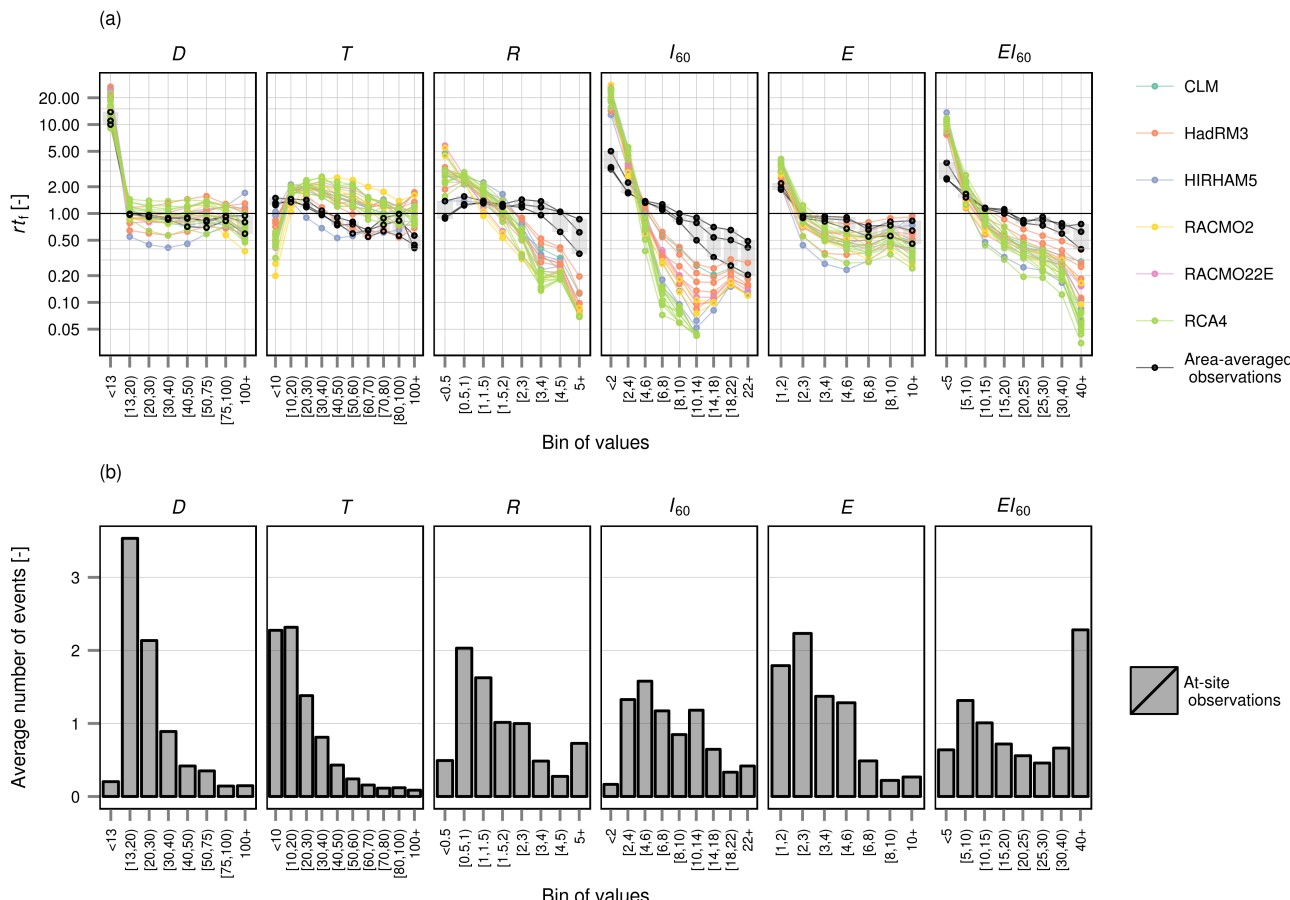

**Figure 5.** (a) Histogram ratios $rt_f$ between areal and station average of frequencies of corresponding bins of the histograms of rainfall event characteristics. Frequencies of bins are averaged per individual season and grid box for the RCM simulations (or neighbourhoods for area-average observations). Grey bars highlight ranges between the 12.5, 25, and 50 km area-average observations (black dots). (b) Histograms of the number of at-site events averaged per individual season and individual station. Units of characteristics defining bins of values are: mm ($D$), h ($T$), mm h$^{-1}$ ($R$, $I_{60}$), MJ ha$^{-1}$ ($E$), and MJ mm ha$^{-1}$ h$^{-1}$ ($EI_{60}$).





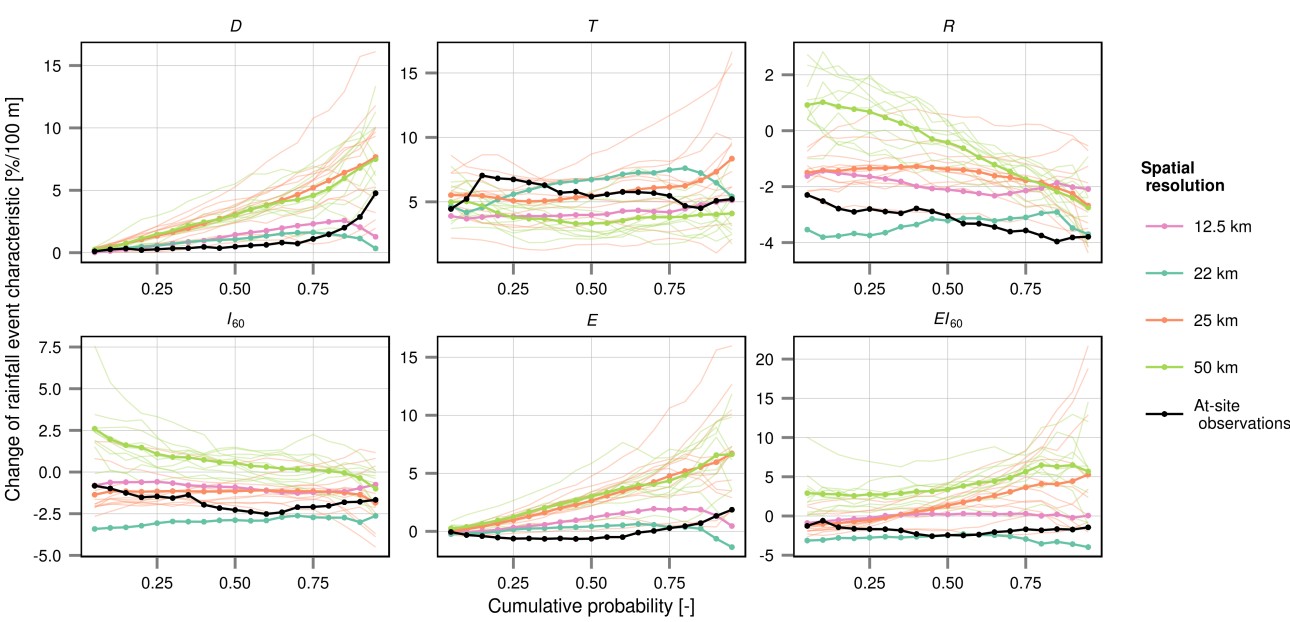

**Figure 6.** Dependence between rainfall event characteristics and altitude. Dependence is expressed by the change of characteristic per 100 m in elevation for the corresponding probability from the empirical distribution function of the event characteristics. Spatial resolution of the RCM simulations is indicated by colour (lines with points show RCM simulations average).