# Peer review of "Characteristics of rainfall events in RCM simulations for the Czech Republic"

_Hydrology and Earth System Sciences, 2016_

## Referee Comment (RC1) · Anonymous Referee #1 · 15 Sep 2016

General comments

The paper evaluates the performance of an ensemble of regional climate models (RCMs) for simulation of heavy rainfall events for the Czech Republic. In recent years, a large number of papers have performed similar analyses of the performance of RCMs by (i) considering different RCMs with different spatial resolutions, (ii) studying different regions of the world, and (iii) analysing a range of rainfall characteristics.

Much of this work has been referred to and discussed in the paper. However, the purpose of the current study and its contribution compared to state-of-the-art in this area is not clear. The authors state that "Characteristics of individual heavy rainfall events (such as event depth, duration, and intensity) in RCM simulations have not been studied to date" (Page 2, line 25-26). The authors may define rainfall characteristics

differently than what has been done in other studies, but the novelty of this compared to the range of other rainfall characteristics that have been analysed in similar studies seems limited. This is confirmed by the fact that the main findings of the study do not provide any convincing new insights. In the Discussion (Section 5.3) the authors discuss their results and compare with previous studies. Basically, all their findings are in agreement with previous studies.

Detailed comments

The results are not presented in a clear and concise way. The Results section provides a detailed enumeration of results shown in the figures. This part should be more to the point and focus on main results.

---

## Referee Comment (RC2) · Anonymous Referee #2 · 15 Sep 2016

The paper compares the characteristics of heavy rainfall events from 23 RCMs with those of point and area-averaged observations in Czech Republic. The methods used to identify heavy rainfall characteristics and to average point observations are new and able to increase our insight about both simulated and observational rainfall events in the country. In general, the paper is well organized and simple to read but difficult to understand some of the approaches used by the authors. As I think the most interesting part of the paper for non-local readers are the methods used in the present study, I suggest the authors to clarify Sections 3.1 and 3.3. For example, the sentences "Alternatively, the minimum event depth of a heavy rainfall event is chosen such that the number of heavy events is on average the same as for observations (=15 %). This approach is similar to the quantile mapping method used frequently for bias correction." given in Section 3.1 is not clear. Or you might add a figure to section 3.3 showing the

generated neighbourhoods for stations corresponding to at least one of the RCMs.

Minor points: Page 2 line 2: Please add more recent studies such as: Danandeh Mehr, A. and Kahya, E. (2016). Grid-based performance evaluation of GCM-RCM combinations for rainfall reproduction. Theoretical and Applied Climatology (in press) Page 3 line 5: Please provide full expression before the abbreviation a.s.l. Page 3 line 6: Please replace "the 1961-2000 period" with "the period 1961-2000"

---

## Author Comment (AC1) · 28 Sep 2016

Referee #2 general comments:

The paper compares the characteristics of heavy rainfall events from 23 RCMs with those of point and area-averaged observations in Czech Republic. The methods used to identify heavy rainfall characteristics and to average point observations are new and able to increase our insight about both simulated and observational rainfall events in the country. In general, the paper is well organized and simple to read but difficult to understand some of the approaches used by the authors. As I think the most interesting part of the paper for non-local readers are the methods used in the present study, I suggest the authors to clarify Sections 3.1 and 3.3. For example, the sentences "Alternatively, the minimum event depth of a heavy rainfall event is chosen such that

the number of heavy events is on average the same as for observations (=15 %). This approach is similar to the quantile mapping method used frequently for bias correction." given in Section 3.1 is not clear. Or you might add a figure to section 3.3 showing the generated neighbourhoods for stations corresponding to at least one of the RCMs.

Our response to general comments:

We agree that the methods and key findings are more relevant for international readers than the detailed results. Therefore we believe that the manuscript can be improved by revisiting and clarification of the methods description (possibly including a visual illustration of the methods) and also by presentation of results in a more concise way without less important details.

Minor points:

Page 2 line 2: Please add more recent studies such as: Danandeh Mehr, A. and Kahya, E. (2016). Grid-based performance evaluation of GCM-RCM combinations for rainfall reproduction. Theoretical and Applied Climatology (in press) Page 3 line 5: Please provide full expression before the abbreviation a.s.l. Page 3 line 6: Please replace "the 1961-2000 period" with "the period 1961-2000"

Our response to minor points:

Thank you for these suggestions, we will consider them in the revised version of the manuscript.

---

## Editor Comment (EC1) · J. Seibert (Editor) · 5 Oct 2016

Dear reviews and authors, thanks for your contributions on discussing this manuscript. Based on this discussion and my own reading I find that this manuscript potentially could make a valuable contribution, but that the results in the original versions were not presented clear enough to allow assessing the novel aspects in this study. In the revisions, the results need to be clarified and the novel aspects to be emphasized more.

Best regards, Jan Seibert
* * *

---

## Author Response (AR1)

Dear prof. Seibert,

hereby we submit a revised version of our manuscript „Characteristics of rainfall events in RCM simulations for the Czech Republic". The critical remarks of the referees and your suggestions resulted in substantial rewrite of the paper. Specifically we focused on highlighting the novelty of our contribution, clear description of the methods used and drawing the attention more to the methods and general results. We also considerably reduced the detailed description of specific results, especially those that are too focused on the study area. We believe that the paper has been substantially improved and thank you and the referees for comments and suggestions.

Best regards,

**Referee #1**

*General comments*

**The paper evaluates the performance of an ensemble of regional climate models (RCMs) for simulation of heavy rainfall events for the Czech Republic. In recent years, a large number of papers have performed similar analyses of the performance of RCMs by (i) considering different RCMs with different spatial resolutions, (ii) studying different regions of the world, and (iii) analysing a range of rainfall characteristics. Much of this work has been referred to and discussed in the paper. However, the purpose of the current study and its contribution compared to state-of-the-art in this area is not clear. The authors state that "Characteristics of individual heavy rainfall events (such as event depth, duration, and intensity) in RCM simulations have not been studied to date" (Page 2, line 25-26). The authors may define rainfall characteristics differently than what has been done in other studies, but the novelty of this compared to the range of other rainfall characteristics that have been analysed in similar studies seems limited.**

In line with the referee we admit that there has been a plethora of studies concerning the evaluation of various precipitation characteristics in global and regional climate models. The vast majority of these studies, however, focused on daily or longer temporal scales, while sub-daily model performance has received relatively little attention to date (Westra et al., 2014). Particularly, we are not aware of another study that would look in detail into *rainfall event* characteristics in RCMs (which are important with respect to hydrological impacts, as detailed below). The few existing studies on RCM-simulated sub-daily rainfall are typically looking at precipitation maxima in a number of temporal aggregations (Hanel and Buishand, 2010; Gregersen et al., 2013), dependence of RCM performance on its resolution (Prein et al., 2016; Sunyer et al., 2016), diurnal cycle of simulated hourly precipitation (Prein et al., 2015 and references therein) or scaling of precipitation extremes with temperature (Lenderink and van Meijgaard, 2008; Ban et al., 2015). Although some of these results can partly be related to rainfall event characteristics (e.g. performance of an RCM in simulating annual precipitation maxima may be related to that in simulating rainfall event maxima), they cannot address questions such as "is the depth of a simulated rainfall event comparable to that of a real event?", "is the performance influenced by the rainfall depth itself?", "are the simulated events of proper length and rain rate?" etc. The importance of these questions in evaluating climate models has been highlighted e.g. by Westra et al. (2014) who suggested (among other things) to focus on (spatial structure and) temporal evolution of rainfall events and their timing and intermittency.

Moreover, characteristics of rainfall events determine characteristics of various hydrological processes, e.g. overland flow generation and shape of the resulting hydrograph (Singh, 1997), soil moisture dynamics (Wang et al., 2008; He et al., 2012), infiltration (Ran et al., 2012), rainfall erosion (Wischmeier and Smith, 1978), evaporation (Dunkerley, 2008), storm sewer flow rates and direct runoff (Schilling, 1991; Giulianelli et al., 2006). Therefore information on changes in these characteristics is highly relevant for river basin management, urban hydrology, flood protection, erosion control etc. That is why we consider these characteristics important also in RCM evaluation studies.

Assessments of simulated precipitation at sub-daily time scales often face problems with data availability – sub-daily RCM simulations are in general not easily available and relevant (sub-daily) observational products allowing for comparison between RCMs (representing spatial averages) and observations (point measurements) are lacking. Deflation of maxima due to spatial averaging is well recognized and expressed by so-called areal reduction factors (Svensson and Jones, 2010). Clearly the spatial averaging also affects the rainfall event characteristics, however, any quantitative assessment is lacking in the literature.

The novelty of our study can be summarized in the following points:

- evaluation of simulated sub-daily precipitation using a large ensemble of RCM simulations from the ENSEMBLES and CORDEX projects
- development of a methodology allowing for comparison of RCM-simulated rain event (spatial) characteristics to (point) observations
- assessment of rainfall event characteristics, such as event depth, event duration, event rain rate, event maximum intensity and indices of rainfall erosivity, which are only indirectly related to commonly

considered indices and are relevant for river basin management, urban hydrology, flood protection, erosion control etc.

We agree that these points could be stated more explicitly in the manuscript, and therefore we elaborated them in the revised manuscript in detail.

**This is confirmed by the fact that the main findings of the study do not provide any convincing new insights. In the Discussion (Section 5.3) the authors discuss their results and compare with previous studies. Basically, all their findings are in agreement with previous studies.**

Our results are indeed consistent with previous studies, however, we also address points that have not been studied before, e.g. the frequency of heavy rainfall events within a year and the skill of climate models in representing the basic characteristics of temporal structure of sub-daily rainfall in particular. We agree that the discussion currently included in the manuscript is to a large extent focused on comparison with other studies instead of addressing the possible consequences for hydrological modelling and climate change impact assessment at short temporal scales in general. Such discussion is supplemented in the revised manuscript.

*Detailed comments*

**The results are not presented in a clear and concise way. The Results section provides a detailed enumeration of results shown in the figures. This part should be more to the point and focus on main results.**

In agreement with this point as well as concerns of Referee #2 we reduced the result section and clarified the methods description.

**Referee #2**

*General comments*

**The paper compares the characteristics of heavy rainfall events from 23 RCMs with those of point and area-averaged observations in Czech Republic. The methods used to identify heavy rainfall characteristics and to average point observations are new and able to increase our insight about both simulated and observational rainfall events in the country. In general, the paper is well organized and simple to read but difficult to understand some of the approaches used by the authors.**

**As I think the most interesting part of the paper for non-local readers are the methods used in the present study, I suggest the authors to clarify Sections 3.1 and 3.3. For example, the sentences "Alternatively, the minimum event depth of a heavy rainfall event is chosen such that the number of heavy events is on average the same as for observations (=15 %). This approach is similar to the quantile mapping method used frequently for bias correction." given in Section 3.1 is not clear. Or you might add a figure to section 3.3 showing the generated neighbourhoods for stations corresponding to at least one of the RCMs.**

We agree that the methods and key findings are more relevant for international readers than the detailed results. Therefore we revised the manuscript with focus on highlighting the novelty of our contribution, clear description of the methods used and drawing the attention more to the methods and general results. We also considerably reduced the detailed description of specific results, especially those that are too focused on the study area.

*Minor points*

**Page 2 line 2: Please add more recent studies such as: Danandeh Mehr, A. and Kahya, E. (2016). Grid-based performance evaluation of GCM-RCM combinations for rainfall reproduction. Theoretical and Applied Climatology (in press)**
**Page 3 line 5: Please provide full expression before the abbreviation a.s.l.**
**Page 3 line 6: Please replace "the 1961-2000 period" with "the period 1961-2000"**

This was corrected.

[revised manuscript text omitted]

---

## Author Response (AR2)

**Report #1**

**Anonymous Referee #3**

*Suggestions for revision*

The paper „Characteristics of rainfall events in RCM simulations for the Czech Republic" by Svoboda et al., has been reviewed by two referees and revised by the authors. The revisions have (a) focussed the message, (b) sharpened the complementarity to the rich literature on RCM simulations of rainfall, and (c) highlighted the general value of the results for an audience beyond the Czech Republic. These were in my opinion the three main concerns and suggestions of the referees. However I have some comments/opinions for discussion, and some issues that may still need to be addressed/clarified before final publication.

**(1) It is true that many RCM studies on precipitation extremes exist and the authors cite some of this work. It is true that many of these analyses are also event-based, so there is limited novelty in that alone (Referee 1). However, I agree with the authors that most published RCM studies did not focus on sub-daily (e.g. hourly) rainfall, simply because the required RCM runs are rather new. I also agree with the authors that there is much to be said for event-based analysis from high-resolution rainfall because it is only with this kind of data/observations that more subtle and important details of the precipitation regime (duration and number of events, event intensity fluctuations, extremes) can be gleaned. I find the novelty of the analysis more clearly stated in the revised manuscript, being the analysis of RCM hourly-scale simulated event characteristics, and the comparison of RCM and point data, i.e. the problem of spatial averaging.**

*Thank you for the comment. We are happy to see that we succeeded in communicating the motivation and novel aspects of our research better.*

**(2) In the introduction the authors state that "Although growing attention has been given to studies at sub-daily time scales in recent years, the complexity of physical processes related to sub-daily extremes and their simplification within climate model parameterizations might discourage researchers from assessment of simulated sub-daily precipitation, particularly since its validation is impaired by the lack of long and high-quality observed rainfall data series at hourly or sub-hourly time scales." This statement seems to suggest that because RCMs cannot simulating hourly rainfall accurately and we have few hourly observation datasets, scientists are discouraged to compare RCMs with observations. I don't think this reasoning is true. First of all there are enough long (20-30 yr) hourly records world-wide to do such comparisons. Secondly, I doubt climate modellers would agree that they are discouraged to compare their simulations with data at hourly resolutions, on the contrary. Please consider revising this or explain better what you mean. Also, the statement that only a few studies have dealt with characteristics of individual rainfall events (page 3, line 17) is not true, the authors themselves contradict this by the extensive literature that exists on defining independent storms from rainfall data (Section 3.1).**

*The point was that because of the lack of detailed validation of RCM simulations at sub-daily time scales (partly caused by the lack of long-term observed precipitation records with sufficient spatial coverage allowing for comparison of observed and RCM-simulated data) researchers might be discouraged from using sub-daily simulated data for the assessment of possible future changes in rainfall event characteristics. The sentence has been rephrased.*

*We agree that the statement that only a few studies have dealt with characteristics of individual rainfall events was unfortunate and therefore it has been deleted.*

**(3) In the RCM simulations chapter additional clarification is needed, e.g. (a) write explicitely what is the time resolution of the individual model outputs (Table 1), I understand only very few RCMs in ENSEMBLES generated sub-daily rainfall, was it always hourly rainfall?; (b) explain what is the sensitivity to external forcing in HadCM3, this is not clear to non-modellers; (c) state explicitly what are the emission scenarios studied.**

*(a) time resolution of the original RCM data has been added to Table 1 and is now mentioned in Sect. 2.3. To our knowledge, most of the ENSEMBLES RCMs did generate sub-daily rainfall (one of the standard output variables available is the daily value of hourly rainfall maxima) but the data are not always archived or they are available only on request.*

*(b) an explanation has been added*

*(c) we considered the historical simulations, i.e. no emission/concentration scenarios were employed, this is now stated explicitly in Sect. 2.3*

**(4) Defining rainfall events based on MIT and independence is pretty straightforward theoretically -- the work of Restrepo-Posada and Eagleson (J. Hydrol., 1982) is a key reference for this. The ad-hoc basis following USLE mentioned on page 5, line 28, is not clear to me. I do not find the reference to Wischmeier and Smith (1978) at all relevant to the identification of independent strom events. Please explain or remove this citation. The authors then analyze only the 15% largest events, and this frequency was determined from observations exceeding 12.7 mm. Is this what gives Nse? I believe this is not explicitly said anywhere. This fraction was also extracted from the RCM simulations as well. In principle I think this is a good choice, because of the bias in event number in RCM simulations in general, however that this analysis refers only to the 15% largest events should be stated more prominently in the abstract, introduction and conclusions.**

*Reference to work of Restrepo-Posada and Eagleson has been added and we have also edited the first two paragraphs of Sect. 3.1. We agree that USLE is not relevant for the determination of independent events, however, the published studies on rainfall events often do not estimate optimal MIT (Dunkerley, 2008) and rely on recommendation, e.g. by USLE (MIT = 6 hours).*

*The information that analysis refers only to the 15% largest events has been emphasized in the abstract, introduction and conclusions.*

**(5) The method to estimate and compare the areal-average rainfall of observations and RCM grids is in my opinion clearly explained in Section 3.3 (Referee 2). The key Fig 3 shows the ratios of RCM simulations to station-averages for three sampling area sizes. The conclusion is that on the average for the 15% largest events, RCMs produce more and longer events, with lower mean depths, and therefore much lower mean intensities than spatially average rain from stations. This applies also to the extremes on the average. Interestingly the very largest quantiles of rain intensity R are grossly underestimated in RCM simulations (Fig 4) and this likely affects the shorter convective events. This leads me to the underlying question behind this paper – RCM simulations of (heavy) rain are not ready to be used for extreme event analysis without some correction. The authors in the discussion claim that bias correction is not sufficient to solve this problem, and refer to other approaches such as the delta change approach by Sorup et al. (2016). Please expand this opinion with more substantiated arguments or statements why exactly the delta change approach is better.**

*Our point in the discussion is that standard (e.g. quantile based) bias correction is not able to correct for temporal structure of rainfall. There already exist advanced methods allowing for correction of temporal and spatial dependence structure (e.g. Mehrorta and Sharma, 2016), their performance for sub-daily precipitation data is however not clear. The advantage of the delta change approach is that it preserves the temporal structure of the observed data. On the contrary it cannot easily accommodate the changes in the temporal structure of rainfall. We have slightly modified the conclusions.*

**(6) The altitude relationships shown in Fig 6 are rather confusing to interpret, i.e. are the apparently stronger altitude relations in RCMs actually related to the spatial distribution of simulated rain. Is it possible that the same elevations in different parts of the country have completely different climatologies? What do we learn from this analysis about the spatial distribution of rain is not clear to me. Also the authors stress that the biggest problem in RCM simulated data with elevation is in the frequency of heavy rain events and the seaosonal totals (bottom of page 13), yet these poor results are not shown, even though they are mentioned in the conclusions. Altogether this part seems detached**

**from the rest of the work.**

*We believe that proper representation of relations between climatological characteristics and altitude is of key importance for simulating climate of Europe and its possible changes and therefore we did not exclude these results. Part of the identified discrepancies can be related to the spatial resolution of the RCM simulations suggesting e.g. that the RCMs with 50 km resolution are not capable to represent the altitude-dependence for any characteristics, while the RCMs with finer resolutions are performing reasonably well in many cases. It is also interesting that the performance of the CLM simulation (resolution 22 km) is comparable to that of the RACMO22E simulation (resolution 12.5 km).*

*The results for number of heavy events ($N_{se}$) and seasonal totals ($S_{se}$) are not shown in Fig. 6, however, the slopes of the regression with altitude are given in text. We prefer not to show the results for $N_{se}$ and $S_{se}$ graphically since these would represent single points in the panels of Fig. 6, which might be confusing.*

**(7) In summary, the paper presents a rather pessimistic picture of the use of RCMs without correction for rainfall. I personally would like to read also a more optimistic opinion of the authors on where do we go for here and what are the ways in which RCM rainfall is usable and/or can be improved in the future.**

*In many applications the RCM-simulated precipitation and temperature cannot be used without correction even at longer scales. Our intention was not to depress climate modellers or researchers using the climate model outputs for climate change impact assessment but to stress the need for improvements in representation of sub-daily rainfall in climate models (either in convection-permitting models or within process parameterizations) and to raise caution against naive use of sub-daily RCM-simulated data for climate change assessment. The identification of weak points is the first step towards improvement. This has been better formulated in the conclusions.*

**Report #2**

**Anonymous Referee #1**

*Suggestions for revision*

My comments have, in general, been adequately addressed in the revised version. I have few additional comments and technical corrections:

**1. Page 2, line 10-19. Very long sentence. I suggest to split: ", while studies" -> ". Studies"**

*This was done.*

**2. Page 5, line 23. "be to many" -> "be too many".**

*The sentence was corrected.*

**3. Page 7, line 17-18. The at-site characteristics are averaged over the entire country before calculation of the ratios. Is this a sound approach? I expect there is a large spatial variability of the characteristics.**

*We agree that the comparison would be more informative if we could compare whole spatial fields of precipitation characteristics, however, due to considerable proportion of grid boxes not including any station we preferred to compare the areal averages in our analysis. The spatial variability is briefly addressed at the end of Section 4.2 (by comparing the coefficient of variation) and Sect. 4.5 (altitude-dependence). For future research, it could be an option to consider radar data for comparison to RCM simulations, provided sufficiently long and homogeneous records exist.*

*In addition to arguments above, it is a common practice in the scientific literature on RCM assessment to average the results over grid boxes for presentation purposes.*

**4. Page 8, line 13-14. "right part of Table 2" -> "top 4 rows in Table 2".**

*This was corrected.*

**5. Page 8, line 20. What is the meaning of "the intensity of the averaging effect"? I would delete "the intensity".**

*We agree that this was confusing, the sentence has been modified as suggested.*

**6. Page 9, line 23. There is some dependence on events depth, but small. I suggest to rephrase: "does not depend on event depth" -> "depends only slightly on event depth".**

*This was done.*

**7. Page 13, line 10. Delete "however".**

*Deleted.*

**8. Page 13, line 23-24. To analyse the impact of the drizzling effect on the results, larger thresholds for defining dry hours could have been used.**

*Please see the response to the next point.*

**9. Page 13, line 29-35. This section is not that clear. Do you just shorten the duration and divide the same depth with that duration? Then, obviously you'll get a larger intensity. A more sound approach would be to analyse the impact on the characteristics by increasing the threshold for defining dry**

**hours. Also, it is not clear why the maximum 60 min intensity will be increased by a shorter duration. Since you have 1-hour data, the maximum 60 min intensity should not be affected.**

*Proportional shortening of events was performed at hourly interval basis, i.e. shortened rainfall event has the same number of intervals, but these intervals are shorter than 1 hour, while the depths are unchanged. The depths from these shorter rainfall intervals were extrapolated back to the hourly scale. Thus we got rainfall event with the same depth, but shorter duration and different other characteristics.*

*Please note that using larger dry-hour threshold results in shorter lengths, larger intensities but also smaller depths (which are represented relatively well). In the beginning of our analysis we experimented also with the 0.3 mm dry-hour threshold for some RCM simulations, but the resulting bias was not much different when the same (0.3 mm) threshold was applied to the observed data. Therefore finally we used the value reported in the literature (0.1 mm). We also considered a quantile based dry-hour threshold (similarly to the definition of minimum event depth), but this led to severe distortion of distribution of some rainfall event characteristics for a number of RCM simulations.*

**10. Page 15, line 21-22. Second sentence says the same as the first sentence.**

*The second sentence was deleted.*

**References**

[revised manuscript text omitted]